# Cryo-EM structure of cadmium-bound human ABCB6
Seung Hun Choi [1], Sang Soo Lee[1], Hyeon You Lee[1], Subin Kim[1], Ji Won Kim[2] & Mi Sun Jin [1] ✉

ATP-binding cassette transporter B6 (ABCB6), a protein essential for heme biosynthesis in mitochondria, also functions as a heavy metal efflux pump. Here, we present cryo-electron microscopy structures of human ABCB6 bound to a cadmium Cd(II) ion in the presence of antioxidant thiol peptides glutathione (GSH) and phytochelatin 2 (PC2) at resolutions of 3.2 and 3.1 Å, respectively. The overall folding of the two structures resembles the inward-facing apo state but with less separation between the two halves of the transporter. Two GSH molecules are symmetrically bound to the Cd(II) ion in a bent conformation, with the central cysteine protruding towards the metal. The N-terminal glutamate and C-terminal glycine of GSH do not directly interact with Cd(II) but contribute to neutralizing positive charges of the binding cavity by forming hydrogen bonds and van der Waals interactions with nearby residues. In the presence of PC2, Cd(II) binding to ABCB6 is similar to that observed with GSH, except that two cysteine residues of each PC2 molecule participate in Cd(II) coordination to form a tetrathiolate. Structural comparison of human ABCB6 and its homologous Atm-type transporters indicate that their distinct substrate specificity might be attributed to variations in the capping residues situated at the top of the substrate-binding cavity.

ATP-binding cassette transporter B6 (ABCB6) is a homodimeric Type IV half-transporter comprising a single transmembrane domain (TMD) and a nucleotide-binding domain (NBD) with an N-terminal extension (TMD0)[1]. The TMD is responsible for substrate recognition and translocation, while the NBD is involved in ATP binding and hydrolysis, providing the energy required for substrate transport. The functional role of TMD0 in ABCB6 is not yet fully understood, but studies indicate that post-translational modifications, such as N-glycosylation at Asn-6 and disulfide bond formation between Cys-8 and Cys-26, occur within the TMD0 region[2,3]. This suggests that TMD0 may perform a regulatory role in the trafficking, localization, or stability of the transporter. ABCB6 was initially identified as the transporter located in the outer membrane of mitochondria that is responsible for the translocation of porphyrins from the cytoplasm to the mitochondria for heme biosynthesis[4]. Mutations in the ABCB6 gene are linked to tissue-specific disorders such as ocular coloboma[5], familial pseudohyperkalemia[6], porphyria, and dyschromatosis universalis hereditaria[7]. However, there are conflicting reports regarding the subcellular localization of ABCB6[8–11] and its involvement in diverse physiological processes, such as protection of cells against oxidative stress[12], heavy metal tolerance[13,14], and resistance to various xenobiotics[15–20]. These findings indicate that the role of ABCB6 extends beyond its originally identified function in mitochondrial porphyrin transport, highlighting the complexity and multifaceted nature of this transporter[21].

ABCB6 is capable of transporting a broad spectrum of substrates. Since the first report describing the physiological role of ABCB6 as a transporter of porphyrin metabolites[4], several studies have revealed its ability to interact with diverse substrates including heavy metals[13,14,22,23] and organic compounds such as tomatine hydrochloride (responsible for defense against plant pathogens) and vertporfin (a drug used as a sensitizer for photodynamic therapy to remove abnormal blood vessels in the eyes)[15]. In addition, ABCB6 can interact with various anticancer drugs including 7-ethyl-10-hydroxy-camptothecin (SN-38), 5-fluorouracil (5-FU), cisplatin, paclitaxel, doxorubicin, methodrexate, and vincristine[16–20]. Furthermore, ABCB6 functions as a carrier for the Langereis (Lan) antigen, and different mutation types of ABCB6 can result in the absence of the Lan antigen on red blood cells, leading to transfusion incompatibility[24].

Cadmium Cd(II) is a trace metal of concern for environmental pollution and human health due to its cytotoxic and carcinogenic properties[25]. Exposure to Cd(II) can result in a range of detrimental effects including organ failure, increased oxidative stress, and eventual apoptosis of cells. In response, living organisms have developed various mechanisms to cope with cadmium toxicity including transporter-mediated sequestration of Cd(II) into vacuoles and efflux out of cells[26,27]. These mechanisms involve the delivery of Cd(II) either as a free metal cation[28] or in complex with the thiol antioxidant glutathione (GSH; γ-Glu-Cys-Gly) in eukaryotic cells[22] or

[1]School of Life Sciences, GIST, 123 Cheomdangwagi-ro, Buk-gu, Gwangju, Republic of Korea. [2]Department of Life Sciences, POSTECH, 77 Cheongam-Ro, Namgu, Pohang, Republic of Korea. ✉e-mail: misunjin@gist.ac.kr

its derivative phytochelatins (PC; (γ-Glu-Cys)$_n$-Gly, $n = 2–11$) in plant cells[29,30]. By contrast, the role of GSH as a detoxifying agent remains relatively understudied in bacteria, although GSH has been shown to influence cadmium tolerance in *Rhizobium leguminosarum bv. Viciae*[31,32] and resistance to arsenite and mercury in *Escherichia coli*[32]. ABCB6 and its functional homolog Heavy Metal Tolerance factor 1 (HMT-1) are essential for detoxifying heavy metals in various species including human, *Rattus norvegicus*, *Caenorhabditis elegans*, *Drosophila melanogaster*, *Schizosaccharomyces pombe*, and *Chlamydomonas reinhardtii*[23,33–35]. For example, *C. elegans* HMT-1 is known to detoxify cadmium, copper, and arsenite[35], while *Drosophila* HMT-1 provides tolerance specifically to cadmium, but not to mercury and arsenite[36]. Moreover, recent evidence has demonstrated that human ABCB6 rescues the cadmium-sensitive phenotypes observed in *C. elegans* and *S. pombe* that result from defective HMT-1 function[14,22,23].

In the present study, we determined cryo-electron microscopy (cryo-EM) structures of human ABCB6 in complex with a Cd(II) ion, which is coordinated by two molecules of either GSH or PC2, at resolutions of 3.2 and 3.1 Å, respectively. Our structural findings shed light on the molecular mechanisms underlying ABCB6-mediated cadmium efflux. Considered alongside our previous data[37,38], the results reveal that ABCB6 employs diverse strategies to actively transport a broad range of substrates, ranging from small heavy metal cations to large organic compound porphyrins.

## Results

### Binding of cadmium to hABCB6$^{core}$ significantly enhances ATPase activity in the presence of GSH

The crystal structure of Atm1 from *Novosphingobium aromaticivorans*, a bacterial homolog of human ABCB6, revealed that it recognizes mercury in combination with GSH[39]. Similarly, HMT-1 family members, ABCB6 homologs present in plants and fungi, are proposed to transport heavy metals in the form of complexes with PCs into the vacuolysosomal compartment[33,40,41]. However, this PC-dependent pathway in heavy metal detoxification is contentious in certain fungi, with conflicting reports suggesting that HMT-1 proteins may not always transport metals in conjunction with PCs[23,42,43]. To investigate the involvement of ABCB6 in metal detoxification, we conducted in vitro functional assays to determine whether the transporter relies on GSH as a cofactor for this process. To ensure accurate results and avoid potential issues related to partial proteolysis, inherent flexibility, and conformational heterogeneity, we utilized the human ABCB6 core region lacking TMD0 (hABCB6$^{core}$) for our experiments[37]. The results of molybdate assays demonstrated that, in the absence of GSH, none of the tested metals exhibited a stimulatory effect on the ATPase activity of hABCB6$^{core}$ (Fig. 1a). In addition, GSH alone did not stimulate the activity of hABCB6$^{core}$, consistent with the behavior observed for the full-length human ABCB6 protein[44]. However, in the presence of GSH, only Cd(II) significantly increased the ATPase activity of hABCB6$^{core}$ by ~2.4-fold (53 nmol/mg/min) at a saturating concentration of 800 μM, compared with the basal state (22 nmol/mg/min; Fig. 1a, b). According to the Michaelis-Menten kinetic analysis, the apparent $K_m$ for Cd(II):GSH was $108 \pm 28$ μM with a $V_{max}$ of $1029 \pm 423$ nmol/mg/min (or 127 min$^{-1}$). By contrast, the catalytically inactive E752Q mutant displayed little basal activity and no Cd(II):GSH-stimulated ATPase activity, probably due to an inability to hydrolyze ATP (Supplementary Fig. 1). An in vivo cytotoxic assay also demonstrated that overexpression of hABCB6$^{core}$ confers cellular resistance to Cd(II). A dose-dependent increase in Cd(II) toxicity led to the death of *Spodoptera frugiperda* (Sf9) cells infected with recombinant baculoviruses carrying the E752Q mutant at concentrations >15 μM (Fig. 1c). By comparison, cells infected with baculoviruses carrying the hABCB6$^{core}$ gene exhibited enhanced viability, retaining 40% compared with untreated controls, even at a concentration as high as 50 μM.

Binding of Cd(II) to hABCB6$^{core}$ in the presence of GSH was confirmed through microscale thermophoresis experiments[45]. A titration series was prepared using a constant GSH concentration (1 μM) and varying amounts of Cd(II) (0–62.5 mM). The dissociation coefficient was determined to be $K_d = 0.49 \pm 0.05$ mM, which is 60-fold lower than the $K_d$ value observed for

coproporphyrin III (CPIII, 8.2 μM), a well-known substrate of ABCB6 (Fig. 1d, Supplementary Fig. 2). The E752Q mutant exhibited a similar binding affinity for Cd(II):GSH as the wild type, suggesting that the diminished catalytic function does not impede substrate binding. As a negative control, both the Q501A mutant (with impaired ability for Cd(II):GSH binding, see below) and hABCB10$^{core}$ (unrelated to heavy metal transport) exhibited much lower affinity, with $K_d$ values of $1.7 \pm 1.0$ mM and $4.8 \pm 1.8$ mM, respectively. We further explored the Cd(II)-stimulated ATPase activity of hABCB6$^{core}$ by examining its response to different GSH derivatives, including PC2 and PC3, as well as oxidized glutathione (GSSG) and ophthalmic acid (OPT; Supplementary Fig. 3). In addition, we examined the impact of two different reducing agents, 1,4-dithiothreitol (DTT) and Tris(2-carboxyethyl)phosphine (TCEP), on the ATPase activity of the transporter. Interestingly, the stimulatory effect of Cd(II) on hABCB6$^{core}$ activity was also observed when PC2 was present, resembling the response seen with GSH (Fig. 1e). By contrast, other tested agents did not exhibit significant effects. Collectively, considering the extent of activity stimulation, GSH and PC2 were chosen as cofactors to capture the Cd(II)-bound conformation of hABCB6$^{core}$ using single-particle cryo-EM.

### Structure determination

For cryo-EM analysis of Cd(II)-bound hABCB6$^{core}$ in the presence of GSH or PC2, nanodisc-reconstituted protein was incubated with 800 μM CdCl$_2$ and 1 mM GSH (or PC2) at 4 °C for 30 min prior to grid preparation. Cryo-EM datasets comprising 5134 and 5059 movies, respectively, were collected on a 200 kV Talos Arctica microscope and subsequently analyzed using the CryoSPARC cryo-EM rocessing platform[46]. Two-dimensional (2D) class averages revealed various views of particles, displaying an inverted V-shaped architecture in which the two NBDs are separated from one another, while the two TMDs are open to the cytoplasm and closed off at the membrane apex. By combining the best particles, the final three-dimensional (3D) reconstruction yielded maps with a global resolution of 3.2 Å and 3.1 Å for C2 symmetry in the presence of GSH or PC2, respectively (Table 1). Local resolution analysis revealed a higher resolution of ~3 Å in the TMD region compared with the NBDs, where the resolution ranged from 3.5 to 6 Å. The final density map demonstrated sufficient quality to construct a model of the entire protein, except for the extreme N-terminus (residues 205–239) and the C-terminus (residues 827–842). Detailed processing workflows for the two cryo-EM datasets are included in Supplementary Figs. 4–9.

### Overall folds

The structures of hABCB6$^{core}$ in complex with Cd(II) and GSH (or PC2) are shaped like a lassical inward-facing transporter in which the substrate-binding cavity within the TMD region is open towards the cytoplasm (Fig. 2). Superimposition indicates that the two structures are nearly identical, with a Cα root-mean-square deviation (r.m.s.d.) of 0.6 Å (Supplementary Fig. 10). Compared with the previously determined apo and porphyrin-bound states of hABCB6$^{core}$[37,44,47], the most notable difference is the degree of separation between the two halves of the transporter (Fig. 3). Consequently, when calculating the volumes of the cavities using the program CASTp with a probe radius of 3 Å[48], the apo conformation had a cavity volume of ~4883 Å$^3$, while the substrate-bound forms exhibited much smaller cavity volumes ranging from 2312 to 2613 Å$^3$ (Fig. 3a). A similar pattern was observed for the distance between the two NBDs; these were more closely aligned towards each other upon substrate binding (Fig. 3b). Structure comparison also revealed subtle differences in their tilts depending on the bound substrate species. In the apo state, the two NBDs are situated in a roughly parallel position relative to each other when viewed from the cytoplasm (Fig. 3b). The smallest tilt difference of 3.9° is observed between the apo and Cd(II):GSH (or PC2)-bound states, with a clockwise rotation of one NBD with respect to the other NBD. The tilt differences between the apo and porphyrin (CPIII or hemin:GSH)-bound states are approximately 22.6° and 15.4°, respectively, with a clockwise and counterclockwise rotation. These findings suggest that the complex and sophisticated modulation of ABCB6 conformations upon substrate binding allows this single transporter

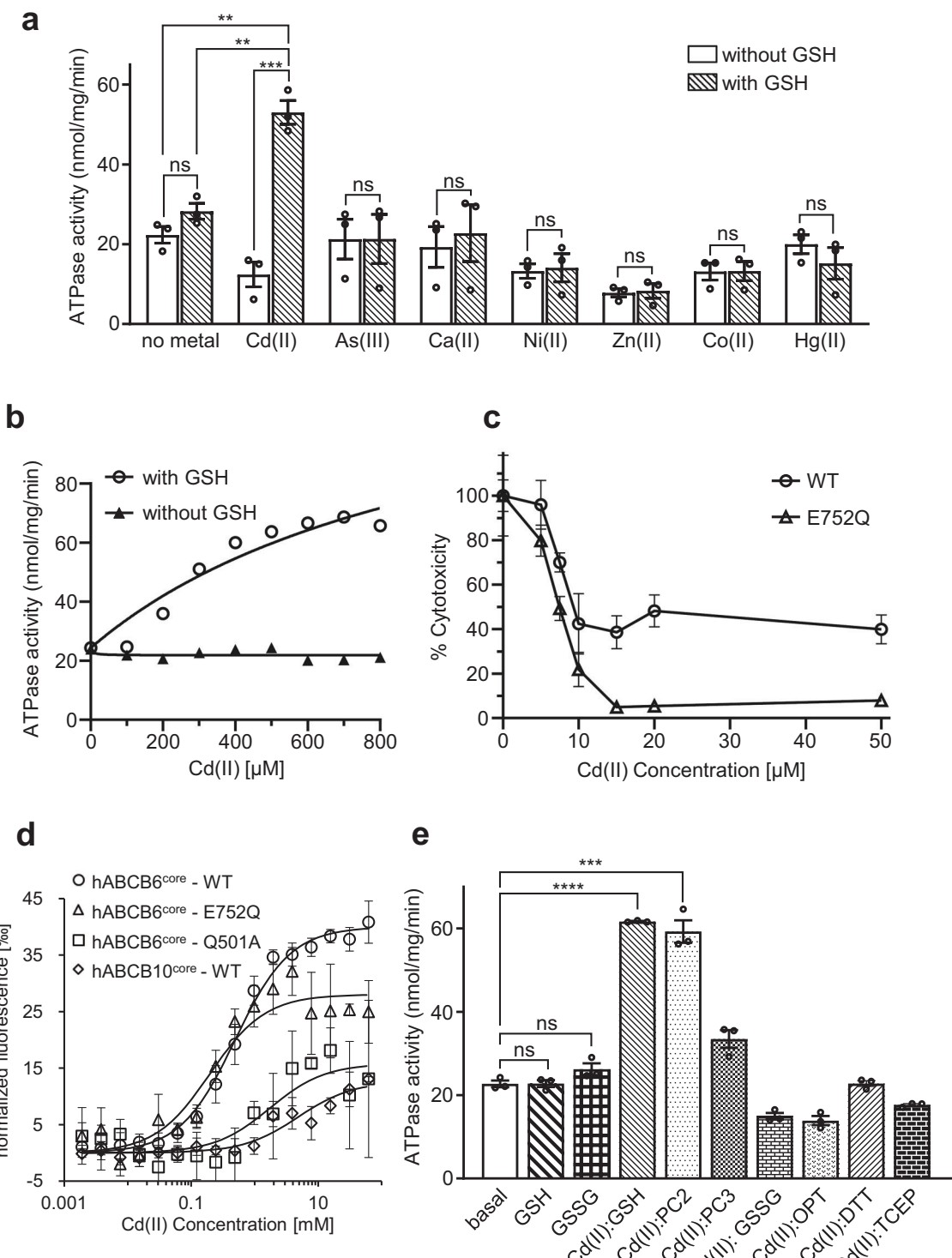

**Fig. 1 | Functional characterization of ABCB6 in mediating cadmium detoxification. a** The ATPase activities of hABCB6$^{core}$ were measured against various metal species at a concentration of 800 μM, in the presence and absence of 1 mM GSH. Values represent the means ± standard error of the mean (SEM) of at least triplicate measurements using two different batches of nanodisc-purified protein. **b** ATPase activity of hABCB6$^{core}$ as a function of Cd(II) concentration with and without 1 mM GSH. **c** Cytotoxicity assay and cell viability profile. Sf9 cells expressing hABCB6$^{core}$ WT or E752Q mutant were cultured with various concentrations of CdCl$_2$ for 2 h. The percentage cell viability was calculated as the ratio of the number of live cells in the presence of Cd(II) to that without Cd(II). **d** Normalized fluorescence changes ($\Delta F_{norm} = F_{hot}/F_{cold}$) were measured to yield binding curves of Cd(II)

to hABCB6$^{core}$ (or its variants) in the presence of GSH. hABCB10$^{core}$ served as a negative control. **e** Cd(II)-stimulated ATPase activity of hABCB6$^{core}$ in the presence of 1 mM GSH or its derivatives. The derivatives tested in the study included phytochelatin 2 (PC2), PC3, oxidized glutathione (GSSG), ophthalmic acid (OPT), 1,4-dithiothreitol (DTT), and Tris(2-carboxyethyl)phosphine (TCEP). GSH (or GSSG) alone-stimulated ATPase activities of hABCB6$^{core}$ were used as negative controls. The symbols **, ***, ****, and ns denote significant differences at $p < 0.01$, $p < 0.001$, $p < 0.0001$, and not statistically significant, respectively, with $p$-values calculated using a two-sided unpaired t-test and adjusted by the Welch's correction method.

**Table 1 | Cryo-EM data collection, refinement and validation statistics**

| | Cd(II):GSH-bound PDB ID 8YR3 EMDB-39534 | Cd(II):PC2-bound PDB ID 8YR4 EMDB-39535 |
|---|---|---|
| *Data collection and processing* | | |
| Magnification | ×100,000 | ×100,000 |
| Voltage (kV) | 200 | 200 |
| Electron exposure (e⁻/Å²) | 50 | 50 |
| Defocus range (μm) | −1.0 ~ −2.2 | −1.0 ~ −2.2 |
| Pixel size (Å) | 0.83 | 0.83 |
| Symmetry imposed | C2 | C2 |
| Initial particle images (no.) | 2,181,111 | 2,007,736 |
| Final particle images (no.) | 402,369 | 398,301 |
| Map resolution (Å) | 3.2 | 3.1 |
| *Refinement* | | |
| Initial model used (PDB code) | 7DNZ | 7DNZ |
| Map sharpening B factor (Å²) | 138.5 | 128.5 |
| *Model composition* | | |
| Protein residues (non-H) | 9319 | 9327 |
| Ligand (non-H) | 41 | 71 |
| *B factors (Å²)* | | |
| Protein | 183.5 | 189.3 |
| Ligand | 174.0 | 196.8 |
| *R.m.s. deviations* | | |
| Bond lengths (Å) | 0.004 | 0.003 |
| Bond angles (°) | 0.679 | 0.679 |
| *Validation* | | |
| MolProbity score | 1.8 | 1.8 |
| Clashscore | 9.7 | 8.4 |
| Poor rotamers (%) | 0.2 | 0.4 |
| *Ramachandran plot* | | |
| Favored (%) | 94.8 | 94.9 |
| Allowed (%) | 5.1 | 5.0 |
| Disallowed (%) | 0.0 | 0.0 |

to recognize a wide range of substrates, spanning from small heavy metals to large porphyrin compounds. This modulation may also contribute to different efficiencies in coupling ATP hydrolysis and transport for these substrates.

## Cd(II) ion-binding site

Along with two molecules of GSH (or PC2) with an orientation perpendicular to the membrane, a Cd(II) ion is positioned within the transmembrane pathway of hABCB6$^{core}$ below TM 7 bulge loop (also known as plug) (Figs. 2 and 4a–d). In the Cd(II):GSH structure, the central cysteine of GSH interacts with the Cd(II) ion, functioning as a major coordination site for metal cation binding. By contrast, the glutamate and glycine residues at the N- and C-terminus of GSH, respectively, are not directly involved in Cd(II) binding. Instead, they form a network of hydrogen bonds and van der Waals interactions with residues within the cavity of the transporter, primarily from TM 9, TM 10, and TM 11 (Fig. 4a, b)[49]. These interactions help to neutralize positive charges within the binding cavity, particularly from the

arginine clusters (R435, R439, and R552), thus facilitating the stable accommodation of a Cd(II) cation. In addition, they contribute to fastening the two halves of the transporter together, ensuring its overall stability and function during the transport process. Consistent with this structural analysis, alanine mutations of the GSH-interacting residues (R435A, R439A, N498A, Q501A, N545A, T549A, and R552A) led to a partial or complete loss of stimulated ATPase activity (Fig. 4e). The mutants R435A, R439A, N498A, and R552A exhibited elevated basal activity compared with hABCB6$^{core}$. This is likely because deletion of these residues induced destabilization of the inward-facing conformation, consequently promoting a transition to the outward-facing conformation even in the absence of Cd(II):GSH.

The overall binding mode of Cd(II) with PC2 in hABCB6$^{core}$ is very similar to that of Cd(II) with GSH, although the molecular basis of complex formation differs (Fig. 4c, d). Specifically, the two cysteine residues of each PC2 molecule participate in Cd(II) coordination to form a tetrathiolate. The glycyl unit at the C-terminus does not participate in interactions with nearby residues and remains freely exposed within the binding cavity. Although human ABCB6 in vivo would not encounter PC2, which is primarily found in plant and certain fungal cells, the high conservation of the mode of complex formation and the residues involved in binding GSH and PC2 (Fig. 4f) suggests that ABCB6/HMT-1/Atm1-like transporters share a common evolutionary origin and mechanism for heavy metal binding and export across diverse organisms. In addition, the overlap between the Cd(II):GSH-binding site in hABCB6$^{core}$ and the site previously identified for hemin:GSH further supports the functional importance of these conserved residues (Supplementary Fig. 11)[37].

The structures determined in this study provide valuable insights into understanding the functional properties of hABCB6$^{core}$. For example, the stable complex formed between Cd(II) and GSH explains why Cd(II) has no stimulatory effect on hABCB6$^{core}$ activity when GSH is absent or replaced by ophthalmic acid, where the cysteine is replaced by 2-aminobutyrate (Fig. 1a, e). The CASTp program calculated that the cavity volumes of the Cd(II)-bound structures in complex with GSH and PC2 to be approximately 2312 Å³ and 2573 Å³, respectively (Fig. 3a)[48]. The volumes of the Cd(II):GSH, PC2 and PC3 complex were estimated to be approximately 649 Å³, 1164 Å³, and 2024 Å³ [50], hence the TMDs may need to move outward to expand the binding cavity and stably accommodate the Cd(II):PC3 complex within it. This alteration may negatively influence the efficiency of ATP hydrolysis, providing a plausible explanation for the low stimulation level of Cd(II) and PC3 on hABCB6$^{core}$ activity (Fig. 1e).

## Structural comparison with the Atm family

Numerous X-ray crystallography and cryo-EM structures have been determined for the Atm1/ABCB7/HMT1/ABCB6-like transporter family bound to their intrinsic substrates, including Atm1 from *Saccharomyces cerevisiae*[51,52], *Novosphingobium aromaticivorans*[39,51,53], and *Chaetomium thermophilum*[54], as well as Atm3 from *Arabidopsis thaliana*[55]. All share a similar overall fold in the core region (Supplementary Figs. 12a–e). Interestingly, *Na*Atm1 is the most different, with a Cα r.m.s.d. of 3.6 Å. These differences primarily stem from TM 4 and TM 5, which are bent and pushed inward towards the center (Supplementary Fig. 12f). However, this difference may potentially arise from distortion caused by non-specific binding of the detergent lauryldimethylamine oxide (LDAO) used during the purification process.

Unlike hABCB6$^{core}$, which does not undergo stimulation by GSH or GSSG alone (Fig. 1e), Atm family members are reportedly responsive to these molecules[39,51,55]. Structural comparison of hABCB6$^{core}$ and Atm proteins reveals that their distinct substrate specificity might be attributed to the different side chains capping the substrate-binding pocket. For instance, in GSSG-bound *At*Atm3[55], the two F435 residues from each monomer are located at the top of the binding cavity, effectively sequestering GSSG within the cavity (Fig. 5a). This residue is replaced by W546 in hABCB6$^{core}$, and its larger side chain may induce steric hindrance with GSSG, which could hinder its binding to the transporter. Furthermore, this structural feature

**Fig. 2 | Overall levels of hABCB6$^{core}$ in complex with Cd(II) and GSH (or PC2). a** Cryo-EM map of Cd(II):GSH-bound hABCB6$^{core}$. The two monomers are colored blue and orange, respectively. The nanodisc is depicted in a gray surface representation. **b** Molecular structure of Cd(II):GSH-bound hABCB6$^{core}$. Cd(II) ion and GSH are shown in green and cyan sphere representation, respectively. The area enlarged in Fig. 4a is boxed. Single primes are used for the residues (or helices) of one monomer to differentiate them from those of the other monomer. **c** Cryo-EM map of Cd(II):PC2-bound hABCB6$^{core}$. The two monomers are colored light blue and light orange, respectively. **d** Molecular structure of Cd(II):PC2-bound hABCB6$^{core}$. The area enlarged in Fig. 4c is boxed.

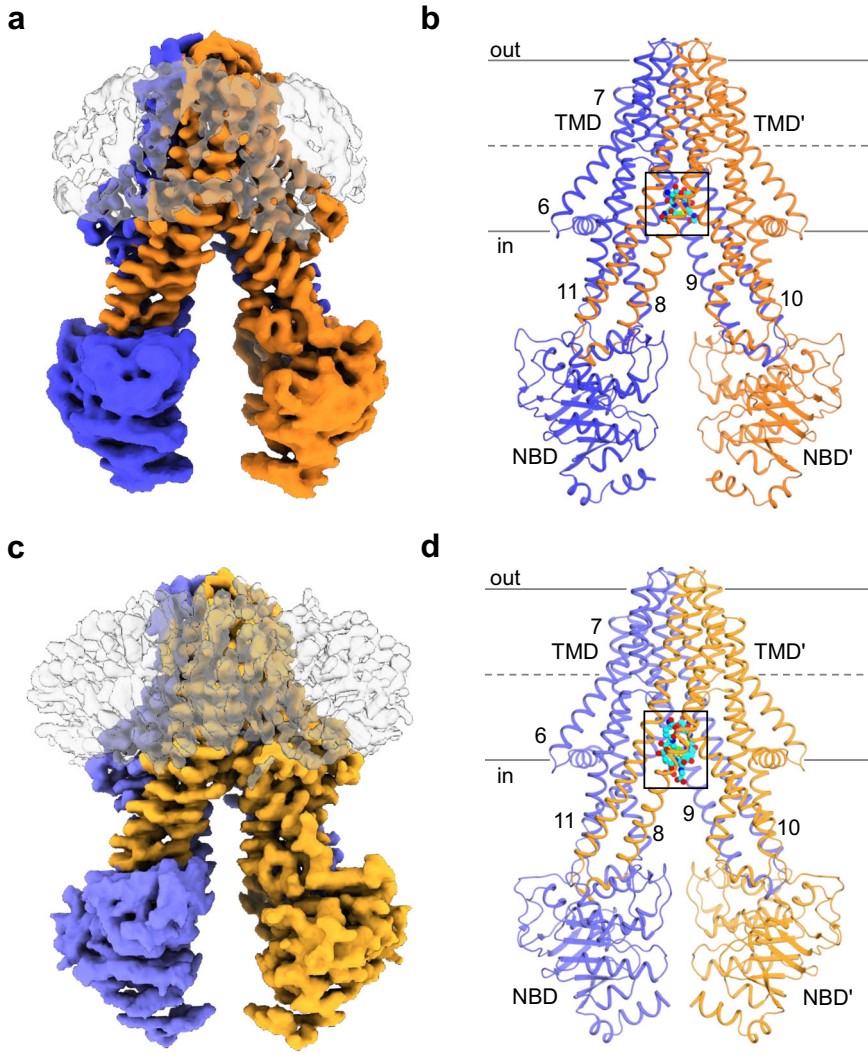

also explains why the Cd(II):GSH-binding site of hABCB6$^{core}$ is shifted slightly downwards by ~2 Å compared with that of the *At*Atm3 transporter, again possibly to avoid steric collision with W546 (Fig. 5a). By comparison, *Na*Atm1 has a methionine residue at position 317 corresponding to W546 in hABCB6$^{core}$ (Fig. 5b–d)[39,53]. The substitution of a relatively small side chain residue in the NaAtm1 cavity creates extra space and shifts the binding site towards the apex, which in turn likely allows for more plasticity when interacting with GSH and related compounds (Fig. 5b–d).

## Discussion

In our study, the ability of the Cd(II):GSH complex to stimulate the ATPase activity of hABCB6$^{core}$ is consistent with previous reports[14,22,23], whereas the lack of stimulation we observed with arsenite As(III) differs[13]. This apparent discrepancy may be attributed to differences in experimental conditions, such as construct design (i.e., full-length protein vs. hABCB6$^{core}$ employed in the present study), and the utilization of different assay systems in vivo (or at the cellular level) vs. in vitro studies. Wang et al. reported that ABCB6 is involved in GSH-dependent copper Cu(II) detoxification based on their colorimetric malachite green assay results[44]. When we attempted to repeat the experiment using the same assay system, we obtained similar results. However, the observation of a positive signal even in the absence of the protein suggests the possibility that the Cu(II) signal could be artifactual and not directly related to ABCB6 activity.

Based on the results obtained from ATPase and cytotoxicity assays (Fig. 1), we determined two structures of Cd(II)-bound hABCB6$^{core}$ in complex with GSH or PC2 (Fig. 2). Despite the potential for trace metals to exhibit various oxidation states, we specifically tested their most stable oxidation states in solution. For example, cobalt is a component of vitamin B12, which supports the production of red blood cells[56]. It commonly displays oxidation states of +2 in an aqueous solution, although the +3 oxidation state is more stable in the majority of its complexes, as found in Co-protoporphyrin IX (Co(III)PPIX), an inhibitor of heme oxygenase[57]. By contrast, cadmium has no known biological function in humans, only serving as a zinc substitute at the catalytic site of specific carbonic anhydrases in certain marine diatoms[58]. It primarily exhibits an oxidation state of +2 in most of its compounds and chemical interactions[59], seemingly precluding its involvement in protoporphyrin. Interestingly, ATPase activity of hABCB6$^{core}$ is stimulated by Co(III)PPIX[37], but no such effect was observed with free cobalt ions (Fig. 1a). By contrast, the interaction of hABCB6$^{core}$ with Cd(II) in the presence of GSH elicits a 2.4-fold increase in ATPase activity and provides cellular protection against Cd(II) toxicity (Fig. 1a–d). Our observations extend to the E752Q variant, which exhibits a Cd(II):GSH-binding affinity comparable with that of hABCB6$^{core}$ but lacks the ability to confer cellular protection. In addition, we endeavored to measure the liposome-based transport activity of hABCB6$^{core}$ using Cd(II)-sensitive fluorescent dyes, including Leadmium green, Rhodamine B, and Fura-2. However, despite our extensive efforts, apparent transport activities were not detected. We speculate that this may reflect the potential influence of Cd(II) ions on the stability and/or rigidity of liposomes, consequently impacting the functional activity of hABCB6$^{core}$[60].

**Fig. 3 | Comparison of the substrate-binding cavities and NBDs of hABCB6$^{core}$ in the presence and absence of bound substrate. a** Overall structures and cavities of hABCB6$^{core}$ in the apo state (PDB ID 7EKM), Cd(II):GSH-bound state, Cd(II):PC2-bound state, CPIII-bound state (PDB ID 7DNY), and hemin:GSH-bound state (PDB ID 7DNZ). The substrate-binding cavities are colored gray, blue, orange, wheat, and green, respectively. **b** Cartoon representations of the NBDs in apo afnd substrate-bound states. The Cα distances between the conserved G626 of the Walker A motif and S728 of the signature motif are indicated. The tilt angle of one NBD in the substrate-bound state with respect to the other NBD in the apo state is indicated. Positive tilt angles represent a clockwise rotation, while negative tilt angles indicate a counterclockwise rotation.

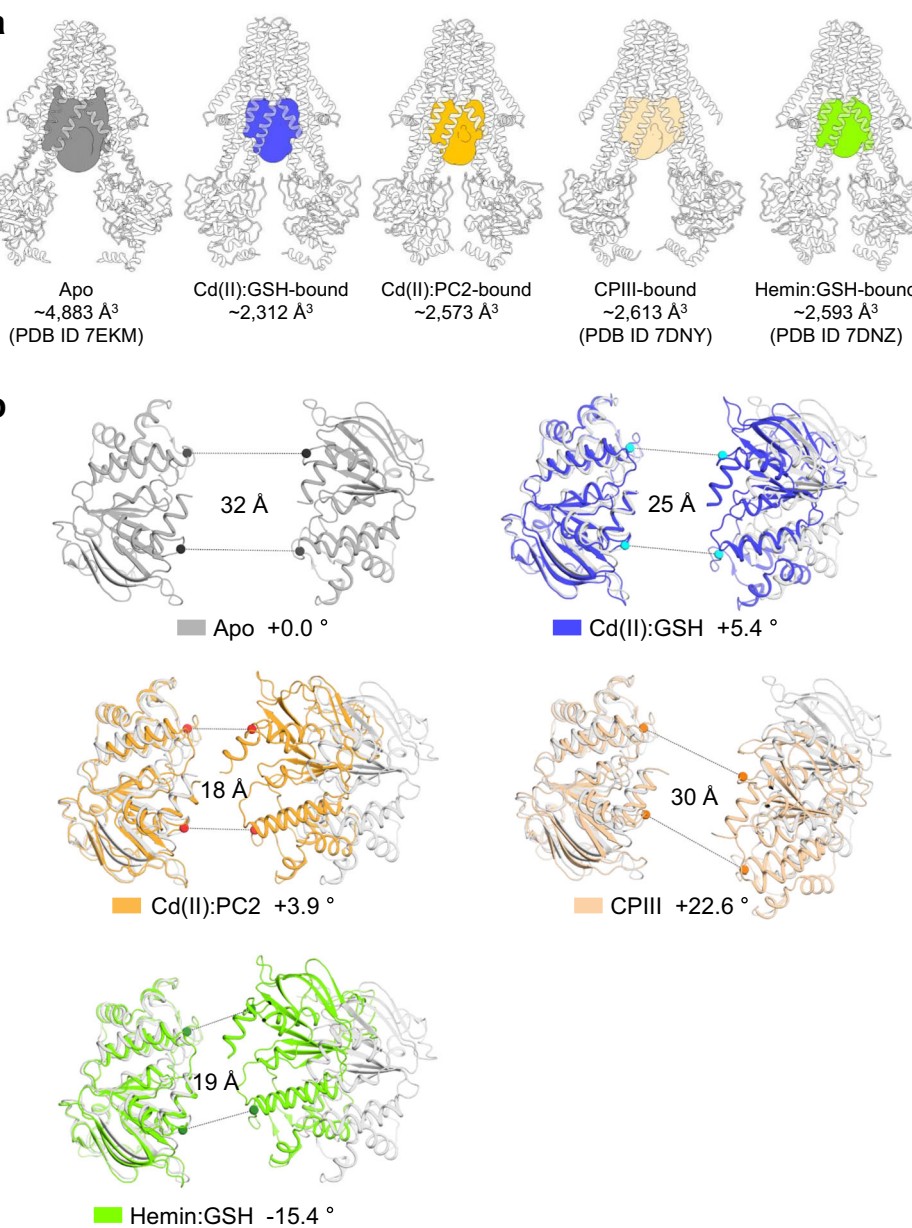

Structural comparison of hABCB6$^{core}$ in three major conformations (apo[47], Cd(II):GSH-bound, and ADP·VO$_4$-bound[38]) unveiled the molecular mechanism of ABCB6-mediated cadmium export. Compared with the apo state, binding of Cd(II):GSH at the interface of the two TMDs brings the TM bundles and NBDs toward the center of the transporter (Fig. 6). Consequently, key residues within the binding pocket move towards Cd(II):GSH, promoting the transporter-substrate interaction (Supplementary Fig. 13), while the TM 7 plug did not exhibit significant conformational change (Supplementary Fig. 14). Subsequent ATP binding and hydrolysis induce NBD closure through large rigid body movements of the TM bundles (Fig. 6). These conformational rearrangements result in adoption of the outward-facing conformation, in which key residues move away from the substrate-binding site and the TM cavity becomes continuous towards the extracellular space. These alterations are likely to decrease the binding affinity of Cd(II):GSH for the hABCB6$^{core}$ and thus facilitate its cellular export. However, in our analysis, the NBD-TMD interfaces remained largely unchanged during the transition from the inward- to the outward-facing conformation, implying that the NBD and the intracellular helical region of the TMD act as a single rigid body during the transport cycle (Supplementary Fig. 15).

Along with previous structural studies[37,38,44,47], our findings offer valuable molecular insights into the enigmatic substrate promiscuity of ABCB6. One notable observation is that ABCB6 exhibits the differential requirement for GSH as a cofactor depending on the substrate type (Fig. 7). When ABCB6 interacts with heavy metals or metal-containing porphyrins, binding of GSH is essential for facilitating their transport. On the other hand, for metal-free porphyrins, GSH is not required for transport. Another important observation is that binding of different substrate species also affects the degree of separation between the two halves of the transporter, as well as the relative orientation between the two NBDs, as observed for human ABCB10 (Figs. 3 and 7)[61,62]. This flexibility and adaptability of the ABCB6 binding cavity and NBDs in response to different substrate species is a key feature that allows ABCB6 to recognize and interact with such a diverse array of substrates.

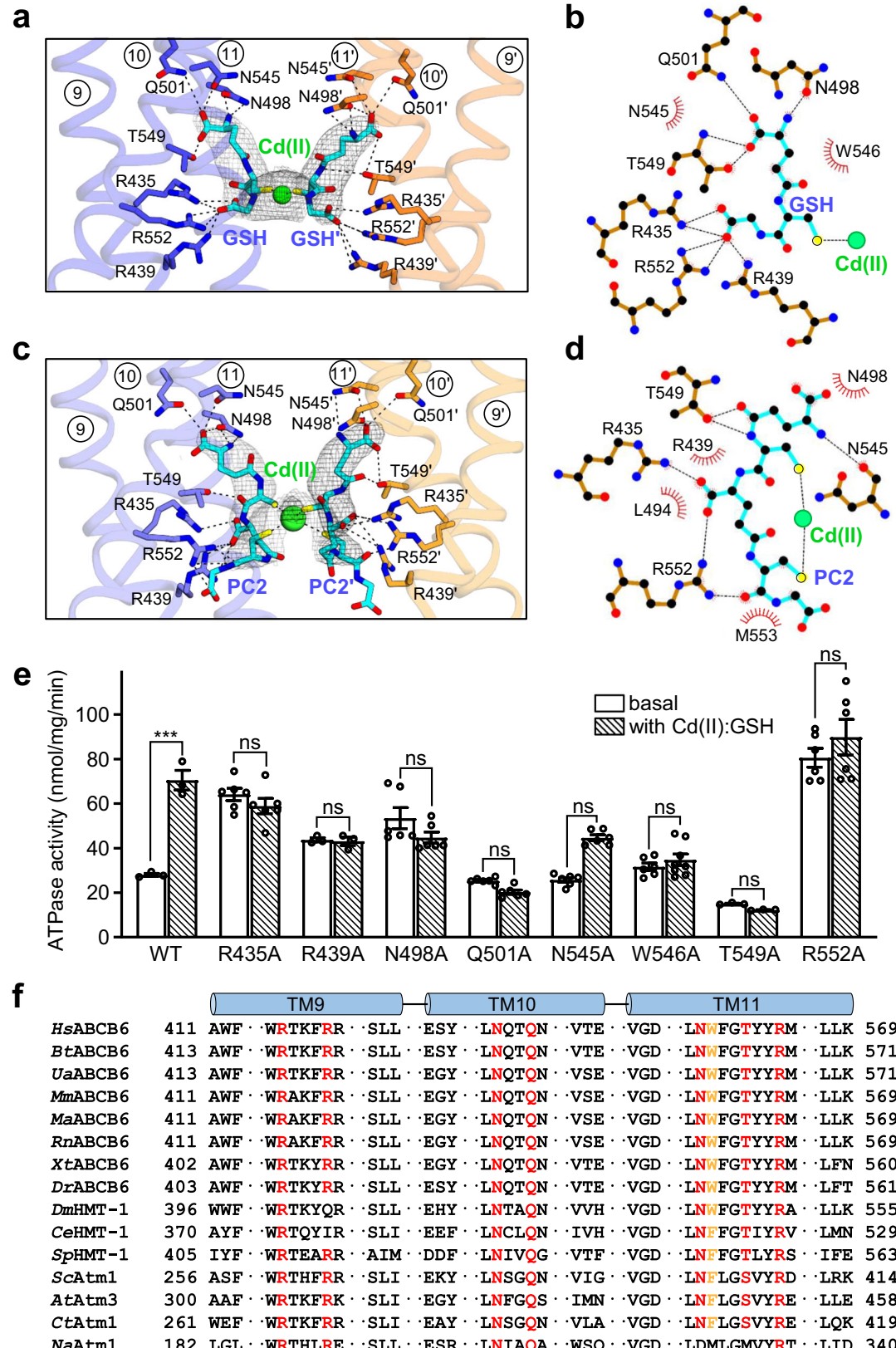

## Methods

### Cloning, protein expression, and purification

hABCB6$^{core}$ and hABCB10$^{core}$ were expressed and purified according to previously described protocols[37,61,62] with minor modifications. Briefly, the codon-optimized synthetic genes of hABCB6$^{core}$ (residues 206–842) and

hABCB10$^{core}$ (residues 152–738) were cloned into the pVL1393 baculovirus transfer vector (BD Biosciences, USA), followed by a thrombin cleavage site, an enhanced green fluorescent protein (eGFP), and a decahistidine (10×His) tag at its C-terminus. Site-directed mutations were introduced via overlap PCR and verified by DNA sequencing. Recombinant plasmids were

Fig. 4 | Close-up view of the substrate-binding site. a Zoom-in view of the Cd(II):GSH-binding site. Bound Cd(II) ion and GSHs are represented by green sphere and cyan stick models, respectively. Cryo-EM maps (gray mesh) of Cd(II) and GSHs are contoured at the 9 and 4 σ level, respectively. The W546 residue is omitted for simplicity (see also Supplementary Fig. 13). The view is rotated by 90 degrees along the vertical axis from Fig. 2b. b Interactions between hABCB6core and the Cd(II):GSH complex analyzed by LigPlot+ software. Residues involved in nonpolar and van der Waals interactions within 4 Å are depicted as red semicircles. c Zoom-in view of bound Cd(II) ion and PC2s. d Interactions between hABCB6core and the Cd(II):PC2 complex analyzed by LigPlot+ software. e ATPase activities of mutants affecting GSH binding. Activity was measured in the presence and absence of 800 μM Cd(II) and 1 mM GSH. Data points represent mean ± standard error of

the mean (SEM) of at least three measurements using two different batches of purified protein. The symbol *** and ns denote significant differences at p < 0.001 and not statistically significant, respectively, with p-values calculated using a two-sided unpaired t-test and Welch's correction. f Sequence alignment of TM helices 9 to 11 of ABCB6 and its orthologs involved in Cd(II):GSH-binding. Highly conserved residues among 15 different species are highlighted in red. Conservation of the capping residues between species is highlighted in orange (see also Fig. 5). Hs, Homo sapiens; Bt, Bos taurus; Ua, Ursus americanus; Mm, Mus musculus; Ma, Mesocricetus auratus; Rn, Rattus norvegicus; Xt, Xenopus tropicalis; Dr, Danio rerio; Dm, Drosophila melanogaster; Ce, Caenorhabditis elegans; Sp, Schizosaccharomyces pombe; Sc, Saccharomyces cerevisiae; At, Arabidopsis thaliana; Ct, Thermochaetoides thermophila; Na, Novosphingobium aromaticivorans.

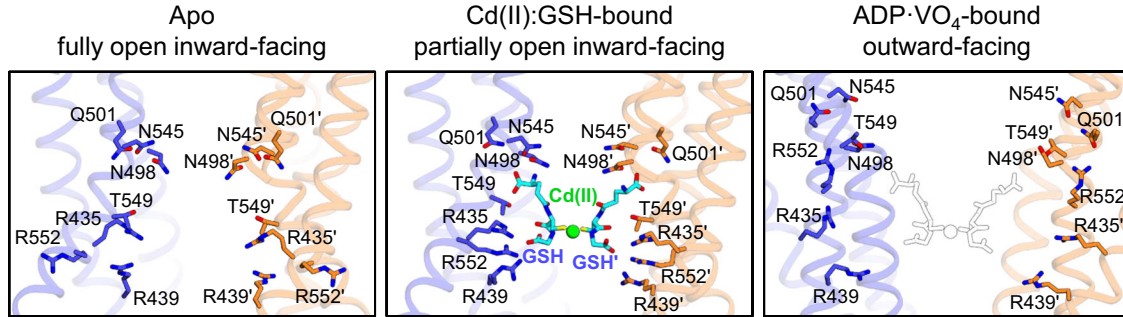

Fig. 5 | Structural comparison of hABCB6core and Atm transporters. a Close-up view of the substrate-binding sites of hABCB6core and AtAtm3 transporters, viewed from within the plane of the membrane. The hABCB6core and AtAtm3 structures are colored blue and light orange, respectively. Bound substrates are depicted as empty black (for the Cd(II):GSH complex in hABCB6core) and orange (for GSSG in

AtAtm3) sticks, respectively. The residue number in parentheses indicates the equivalent residues in AtAtm3. b–d Close-up view of the GSH- (b), GSSG- (c), and mercury Hg(II):GSH- (d) binding sites of NaAtm1 superimposed on the hABCB6core structure. The NaAtm3 structures and the bound substrates are colored light pink and pink, respectively.

then transfected into *Spodoptera frugiperda* (Sf9) cells with BestBac 2.0 linearized baculovirus DNA (Expression Systems, USA) and Cellfectin II transfection reagent (Gibco, USA). The proteins were overexpressed in High Five (Hi5) cells at 28 °C and harvested 72 h post-infection.

For purification of hABCB6core and its mutant proteins, cells were harvested by centrifugation at $10,000 \times g$ for 10 min, followed by resuspension in lysis buffer containing 20 mM HEPES-NaOH pH 7.0, 200 mM NaCl, 5% (w/v) glycerol, 10 mM MgCl$_2$, 40 μg/mL Dnase I, and 0.1 mM phenylmethylsulfonyl fluoride (PMSF; Goldbio, USA). Subsequently, cells were disrupted by sonication using a Branson Sonifier (Emerson, USA) equipped with a 3.2 mm tip at 45% amplitude with 5 s pulses separated by 5 s pauses for 3 min. Cell membranes were collected by ultracentrifugation at $300,000 \times g$ for 1 h. The resulting pellets were solubilized for 2 h using lysis buffer supplemented with 2% (w/v) n-dodecyl-β-D-maltopyranoside (DDM; Anatrace, USA) and 0.2% (w/v) cholesteryl hemisuccinate (CHS; Anatrace). Insoluble cell debris was removed by centrifugation at $300,000 \times g$ for 30 min. The supernatant was then loaded onto anti-GFP DARPin-conjugated agarose resin[63]. After thoroughly washing the resin with 20 column volumes of lysis buffer, the C-terminal tag was removed by on-column thrombin cleavage. Proteins were eluted with buffer containing 20 mM HEPES-NaOH pH 7.0, 200 mM NaCl, 0.056% (w/v) 6-cyclohexyl-1-hexyl-β-D-maltoside (Cymal-6; Anatrace), and 0.0056% (w/v) CHS. Eluted protein was further purified by gel filtration chromatography using a Superdex 200 Increase 10/300 GL column (Cytiva, USA). All purification steps were performed on ice or at 4 °C.

Purification of hABCB10core was conducted following the same procedures employed for hABCB6core, with the only changes being the lysis buffer composition (20 mM Tris-HCl pH 8.0, 100 mM NaCl, 10 mM MgCl$_2$, 40 μg/mL Dnase I, and 0.1 mM PMSF) and the detergent (0.0174% (w/v) DDM and 0.00174% (w/v) CHS) used during gel filtration chromatography.

### Nanodisc reconstruction

Membrane scaffold protein 1 (MSP1), a truncated mutant of human Apo A-I (lacking amino acids 1–43), was used for nanodisc reconstitution[64]. MSP1 was cloned into the pET21a vector (Novagen, Germany) with a thrombin-cleavable hexahistidine (6×His) tag at its C-terminus, expressed in *Escherichia coli* BL21 (DE3) cells in lysogeny broth (LB) medium at 37 °C, and induced with 1 mM isopropyl-β-D-thiogalactopyranoside (IPTG; Goldbio) at OD$_{600}$ for 5 h at 30 °C. The protein was then purified through Ni-NTA affinity chromatography (Goldbio), HiTrap Q anion exchange (Cytiva), and Superdex 200 Increase 10/300 GL gel filtration chromatography.

To initiate nanodisc assembly, porcine brain polar lipid extract (Avanti Polar Lipids, Inc., USA) dissolved in chloroform was dried under a gentle stream of nitrogen. The lipid film was dissolved in buffer containing 20 mM HEPES-NaOH pH 7.0, 200 mM NaCl, 0.056% (w/v) Cymal-6, and 0.0056% (w/v) CHS. Detergent-purified hABCB6core and MSP1 proteins were mixed with the lipid solution at a molar ratio of 1:3:150 and incubated at 4 °C for 2 h. Remaining detergents were removed using Bio-Beads SM-2 resin (Bio-Rad, USA), followed by Superdex 200 Increase 10/300 GL gel filtration chromatography equilibrated with buffer comprising 20 mM HEPES-NaOH pH 7.0 and 200 mM NaCl. Fractions containing the successfully reconstructed protein-nanodisc complex were collected and used for cryo-EM grid preparation and functional studies.

### ATPase assay

The ATPase activities of hABCB6core and its mutants were measured using a molybdate assay as previously described[65]. Briefly, 1.4 μM protein was mixed with 50 μL of reaction buffer containing 50 mM HEPES-KOH pH 7.0, 70 mM KCl, 10 mM MgCl$_2$, 0.056% (w/v) Cymal-6, and 0.0056% (w/v) CHS in the presence or absence of various heavy metals at a final concentration of 800 μM. After incubation for 30 min on ice, the ATP

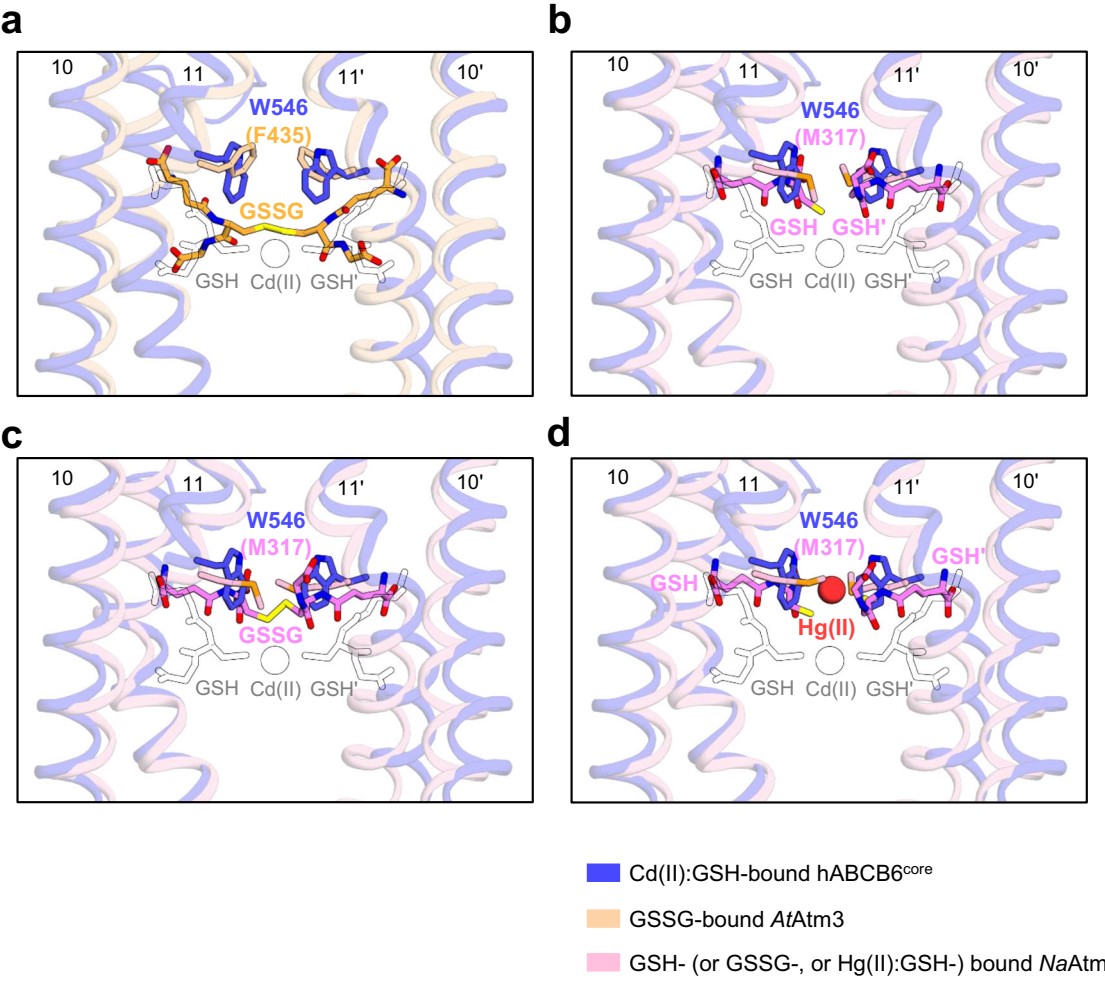

**Fig. 6 | Structural comparison of the substrate-binding sites of hABCB6<sup>core</sup> in apo, Cd(II):GSH-bound, and ADP·VO₄-bound conformations.** Close-up views of the substrate-binding sites of hABCB6<sup>core</sup> in apo (PDB ID 7EKM), Cd(II):GSH-bound, and ADP·VO₄-bound (PDB ID 8KYC) conformations. Key residues interacting with Cd(II):GSH are shown in stick representation. Cd(II):GSH superimposed on the outward-facing hABCB6<sup>core</sup> is depicted as empty black sticks.

hydrolysis reaction was initiated by adding 1 mM ATP, and samples were further incubated for 30 min at 37 °C. To quench the reaction, 50 μL of 10% (w/v) sodium dodecyl sulfate was added and the mixture was incubated for an additional 20 min at 37 °C with a 100 μL solution containing 5 mM ammonium molybdate, 2 mM zinc acetate, and 5.5% (w/v) ascorbic acid. The concentration of Pi released was determined by measuring the absorbance at 850 nm and the ATP hydrolysis activities were calculated using potassium phosphate as a standard for comparison. Curve fitting via non-linear regression analysis was conducted using PRISM 10.0 software (GraphPad, USA).

**Microscale thermophoresis assay**
The binding affinity of Cd(II):GSH to hABCB6<sup>core</sup> (or its variants) and hABCB10<sup>core</sup> was measured by microscale thermophoresis assay as described previously[45]. Briefly, protein was incubated with a 10× molar excess of RED-MALEMIDE 2nd generation in 20 mM HEPES-NaOH pH 7.0 at room temperature (RT) for 30 min. Excess dye was then removed using a desalting column following the manufacturer's instructions (NanoTemper Technologies). Labeled protein (10 nM) was then incubated for 15 min at RT with serially 2-fold diluted Cd(II) (0–62.5 mM) in the presence of 1 μM GSH. Next, the resulting mixtures were loaded onto Monolith premium capillaries (NanoTemper Technologies, Germany). Fluorescence changes, resulting from the thermophoretic movements of the labeled proteins induced by infrared laser activation, were measured using a Monolith

NT.115 pico instrument (NanoTemper Technologies) and plotted against varying concentrations of Cd(II). The results were analyzed using MO.Affinity Analysis software (NanoTemper Technologies).

**Cytotoxicity assay**
Sf9 cells at a density of $150 \times 10^4$ cells per well were attached to a 6-well plate and infected with recombinant baculovirus carrying the hABCB6<sup>core</sup>-eGFP gene. After 48 h post-infection at 28 °C, cells were further incubated for 2 h following addition of various concentrations of CdCl₂ (ranging from 0 to 50 μM) to the medium. Subsequently, dead cells were removed by washing with 2 mL of phosphate-buffered saline (PBS) and the remaining cells were harvested using a cell scraper. The percentage cell viability was calculated as the quantity of live cells in the presence of Cd(II) ions relative to that without Cd(II). Cells infected with baculovirus carrying the E752Q mutant-eGFP gene were also cultured and monitored in the same manner as negative controls. Expression of hABCB6<sup>core</sup> (or E752Q) was monitored using fluorescence microscopy.

**Cryo-EM sample preparation and data acquisition**
Before grid preparation, the nanodisc-purified protein (0.5 mg/mL) was incubated with 800 μM Cd(II) in the presence of 1 mM GSH (or PC2) for 30 min at 4 °C. Next, 3 μL of the protein sample was applied to freshly glow-discharged (15 mA, 30 s) 300-mesh Au R1.2/1.3 holey carbon grids (Quantifoil, Germany). The grids were blotted once for 2–2.5 s with a blot

**Fig. 7 | Schematic diagram of the conformational changes of ABCB6 depending on the bound substrate type.** (i) The need for GSH as a cofactor, (ii) the degree of separation between the two halves of the transporter, and (iii) the tilt angles of the NBDs differ depending on the substrate type bound to ABCB6. The two subunits are colored blue and orange, respectively. The TMD0 domain is omitted for simplicity.

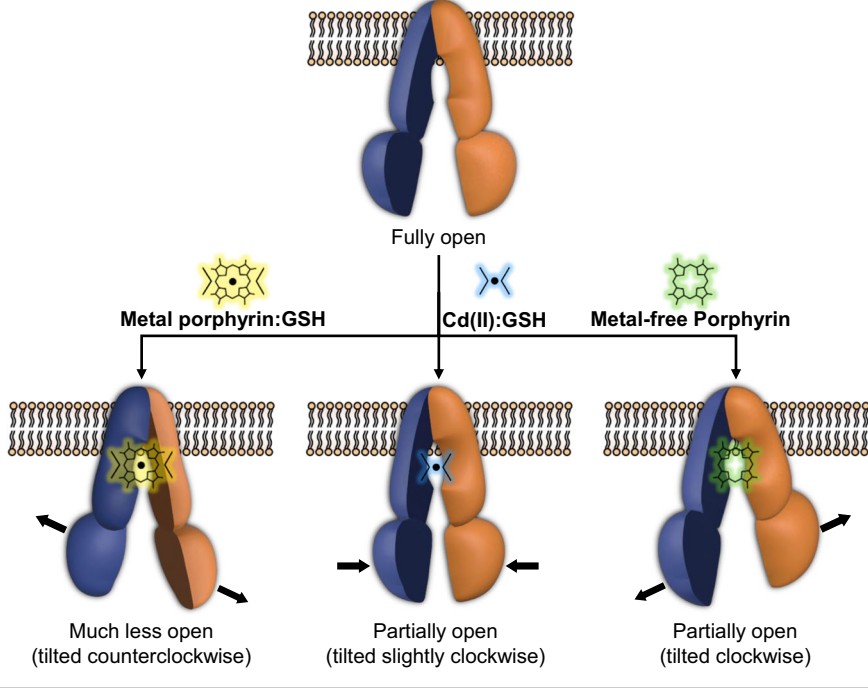

force of 2, then rapidly plunge-frozen in liquid ethane using a Vitrobot Mark IV (Thermo Fisher Scientific, USA) in which the chamber was maintained at 4 °C and 100% humidity. All cryo-EM data were collected using a 200 kV Talos Arctica cryo-transmission electron microscope (Thermo Fisher Scientific) equipped with a K3 BioQuantum direct electron detector and BioQuantum energy filter (20 eV slit width; Gatan, USA). All movie stacks were acquired using EPU software (Thermo Fisher Scientific) in counting mode at ×100,000 magnification with a pixel size of 0.83 Å. Each movie was dose-fractionated to 50 frames with a total exposure time of 3.42 s. The electron dose rate was 10 e$^-$/pix/s (1 e$^-$/Å$^2$/frame). The defocus range was between −1.0 and −2.2 μm with 0.2 steps.

### Cryo-EM data processing
The EM data processing workflows for the Cd(II):GSH- and Cd(II):PC2-bound datasets, consisting of 5134 and 5059 movies, respectively, are presented in Supplementary Figs. 4–9. Cryo-EM data processing was performed using CryoSPARC v4.1.2[46] and a similar strategy was employed for all datasets. Briefly, the beam-induced motion of the movies was corrected via Patch Motion Correction in CryoSPARC. The contrast transfer function (CTF) was calculated by Patch CTF Estimation. Twenty micrographs were selected randomly and particles were picked for 2D template generation using Blob Picker and 2D Classification. Template Picker was then applied to the micrographs and automatically picked particles were extracted using a box size of 280 × 280 pixels. After multiple rounds of reference-free 2D classification, selected particles were used to generate ab initio models. Heterogeneous 3D refinement was conducted for each model and the best class was selected for the final homogeneous 3D refinement with C1 symmetry. All maps were further improved by global and local CTF refinement. Non-uniform refinement[66] and additional 3D classification did not improve map quality. To enhance the quality of the map and local resolution at the Cd(II):GSH (or PC2) binding site, the final map was subjected to post-processing using Resolve Cryo EM in PHENIX[67]. Overall resolutions were estimated based on a gold-standard Fourier Shell Correlation (FSC) cut-off of 0.143 between the two independently-refined half-maps[68–70]. Local resolution was calculated from the two half-maps in cryoSPARC and displayed in UCSF Chimera[71].

### Model building and refinement
The previously reported cryo-EM structure of hABCB6$^{core}$ in complex with hemin:GSH (PDB ID 7DNZ) served as an initial model[37]. The overall structure was manually fitted to the map using UCSF Chimera[71] and subjected to real space refinement using PHENIX with secondary structure, rotamer, Ramachandran, and noncrystallographic symmetry (NCS) restraints[67]. Refined models were built manually in Coot[72] and further refined through iterative rounds of model building in Coot and refinement in PHENIX. Regions with poor density were modeled as polyalanine. The geometry of the final model was validated using MolProbity[73]. All structure representations in the manuscript were generated using UCSF Chimera[71], ChimeraX[74], and PyMOL[75].

### Statistics and reproducibility
Statistical analysis were performed with calculating p-values using a two-sided unpaired t-test and adjusted by the Welch's correction method. The symbols **, ***, ****, and ns denote significant differences at $p < 0.01$, $p < 0.001$, $p < 0.0001$, and not statistically significant, respectively. Error bars in figures represent mean ± standard error of the mean (SEM).

### Reporting summary
Further information on research design is available in the Nature Portfolio Reporting Summary linked to this article.

### Data availability
The refined atomic coordinates of hABCB6$^{core}$ bound to Cd(II) in the presence of GSH or PC2 have been deposited in the Protein Data Bank under accession codes 8YR3 and 8YR4, respectively. The cryo-EM density maps have been deposited in the Electron Microscopy Data Bank under accession codes EMD-39534 (Cd(II):GSH-bound) and EMD-39535 (Cd(II):PC2-bound). The source data behind the graphs in the paper can be found in Supplementary Data 1.

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

## Acknowledgements

We thank Mrs. Su Jeong Kim of POSTECH for helping with EM grid screening and data collection. This research was supported by grants from the National Research Foundation (NRF) funded by the Ministry of Science, ICT, and Future Planning of Korea (NRF-2019M3E5D6063908, NRF-2021M3A9I4022846, and NRF-2022R1A2C1091278), and by a grant from the GIST Research Institute (GRI) IIBR funded by GIST in 2023.

## Author contributions

S.H.C. and M.S.J. conceived the experimental design. S.H.C., S.S.L., and H.Y.L. performed protein purification. S.H.C. conducted biochemical experiments. S.H.C., H.Y.L., and S.K. carried out cryo-EM experiments. Access to the cryo-EM facility was granted by J.W.K. S.H.C. prepared figures and M.S.J. wrote the manuscript with contributions from all authors.

## Competing interests

The authors declare no competing interests.
