## [Peer Review File · Communications Biology]

Reviewers' comments:

Reviewer #1 (Remarks to the Author):

Cryo-EM structure of cadmium-bound human ABCB6

Seung Hun Choi, Sang Soo Lee, Hyeon You Lee, Subin Kim, Ji Won Kim & Mi Sun Jin

The authors presented cryo-electron microscopy structures of core-human ABCB6 (ABCB6 Δ TMD0) bound to a cadmium Cd(II) ion in the presence of antioxidant thiol peptides glutathione (GSH) and phytochelatin 2 (PC2) at resolutions of 3.2 and 3.1 Å, respectively. The overall structure is inward-facing similar to the apo state. Two GSH molecules are symmetrically bound to the Cd(II) ion through their central cysteine, and biochemical assays delineate that presence of GSH associated metal ion effectively stimulates ATPase activity. Degrees of separation between the two halves of the transporter and tilt angles of the NBDs is substrate specific. The manuscript is overall well written and interesting. Core hABCB6 is used in this paper as well as in previous papers, where TMD0 has been deleted i.e., the construct is Δ TMD0.

This group has determined two structures of Cd(II)-bound hABCB6core in complex with GSH or PC2. ABCB6 exhibits the differential requirement for GSH as a cofactor depending on the substrate type. GSH is metal lover. Wherever metal is there either in heavy metal state or porphyrin bound state, GSH facilitates/stimulates hydrolysis thereby resulting in transport.

Concerns:

- (1) No transport assay is shown in this paper; however, transport assay was shown in the previous paper for Co-III only. This should be addressed.
- (2) The degree of separation has a relation with ATPase activity, more the separation less the hydrolytic effectivity? is this correct?
- (3) Previously Co-III-ppIX showed significant stimulation of hydrolysis in presence of GSH. Here Co(II) is used. Which is more biologically relevant? Free heavy metal ions or PPIX associated? This should be addressed.
- (4) Same is for cadmium in this study. Only free Cd(II) and Cd(II)-GSH were utilized. Is higher oxidation state possible here? Is there any possibility of complex formation with PPIX? Any biological relevance? These should be mentioned.
- (5) Why not a structure in presence of non-hydrolysable ATP analog and metal ligands?
- (6) How TM helix 7 is oriented here?
- (7) Fig 1(a)- For ATPase assay, hABCB6 protein concentration (800 μ M) and GSH concentration (1mM). Have you tried for different ratio like 1:2 or more?
- (8) Fig 1(c) - For normalized fluorescence change, what is the incubation time for GSH?
- (9) Fig (f) line 462, what is the significance of having conserved residues in 15 different species? Are these residues important for GSH:Cd(II) only? Not clear in the manuscript.
- (10) All mutants gave partial or no binding except R552. The reasons are not explained properly. ATPase assay is not clear hence needs to be elaborated. ATP binding pocket has not been mentioned here.
- (11) line no.186: 'structural analysis, alanine mutations of the GSH-interacting residues (R435A, R439A, R552A, N498A, Q501A, N545A, and T549A) led to a partial or complete loss of hABCB6 core ATPase activity.' what about the residues of NBD? How does hydrolysis take place? What are the ATP interacting residues? Mutational studies in NBD will improve the manuscript.
- (12) Time scan during measurement of ATPase activity would have been a better choice.
- (13) Authors should discuss the about structural interface of NBD-TMD? Also, the paper <https://doi.org/10.1038/s41467-023-37851-9> should be referred.
- (14) line no: 489- 'the different tilt angles and direction of the NBDs, which can influence the efficiency of ATP hydrolysis and subsequent transport processes, and are affected by the bound substrate species' ---- have they checked this with the other metal ions? Not clear.
- (15) line no.76. 'Our structural findings shed light on the molecular mechanisms underlying ABCB6-mediated cadmium efflux'. The molecular mechanism of Cadmium efflux is not clearly understood.

Reviewer #2 (Remarks to the Author):

In the current study, Choi et al. show that cadmium binds to hABCB6 in a GSH-dependent manner and further determines the structure of cadmium-bound hABCB6 using cryo-EM and an hABCB6 mutant (ABCB6core) that they derived in a previous paper. ABCB6 is an important and current area of research. The protein may be critical for heme synthesis and is implicated in metal transport, transfusion compatibility, mitochondrial metabolism, xenobiotics, and cancer treatment. Despite the importance of ABCB6, its role in these processes is largely unstudied. Specific to this paper, the authors show that ABCB6 may play a role in cadmium toxicity and metabolism. This is an important advancement in the field of toxicology and continues their previous work with hABCB6 in porphyrin binding and transport. The authors do a wonderful job explaining the current research and made stunning figures. That said I would suggest the follow minor and possibly major revisions.

Possible major revision:

-The data set for the ATPase activity of hABCB6core + Cd(II) seems to be the same in figures 1a, 1d, and 4e; yet the negative control data sets (basal and GSH alone) in these figures all seem to be different. Was this a copying mistake? Were the same data sets used in different experiments? If this positive control data set was reused for different experiments, I would suggest that the authors repeat these experiments with the proper controls.

Minor revisions:

-For figures 1a, 1d, and 4e, I would suggest that the authors add statistical significance to their graphs, if any exists.

-It seems like from figure 1a, that GSH does increase the activity of hABCB6core. Was this significant or just a trend? This is different than what is stated in the text. I would suggest that the authors address this. Additionally, there seems to be a discrepancy between figures 1a and 1d. GSH in 1a seems to activate hABCB6core but has no effect in 1d. I would suggest that the authors address this.

Suggestions/Comments:

-Did cadmium in the absence of GSH inhibit hABCB6core activity? If so, I would suggest that the authors make a brief mention of this.

-800uM metal concentration is quite high, and way beyond human physiological levels. Have the authors tried titrating the metals, and specifically the Cd(II) concentration?

-Post-translational-modifications are seemingly very important for ABCB6. As the authors have noted, the precise location of ABCB6 is disputed and mature red blood cells seem to express glycosylated ABCB6 on the plasma membrane which may differ from mitochondrial ABCB6 if it exists in red cells. Given that the recombinant human ABCB6core used in this study is generated from Sf9 (armyworm cells), have the authors checked that the post-translational-modification profile of ABCB6core is similar to ABCB6 found in human cells? The translational-value of these studies would be increased if this were known.

Reviewer #3 (Remarks to the Author):

In this original research paper, Choi et al. report their newest findings on the human ABCB6 transporter. The authors determined the structure of nanodiscs-reconstituted, truncated hABCB6 in complex with Cd(II)-(GSH)₂ and Cd(II)-(phytochelatin)₂ in inward-facing conformations. Comparison of their new complex structures reported here to previously published substrate-bound structures of hABCB6 revealed large flexibility at the NBD region. To further support the physiological relevance of their structural findings, the authors performed in vitro ATPase assays in the absence and presence of substrates and determined the binding affinity of Cd(II)-(GSH)₂ to ABCB6 using microscale thermophoresis. Based on these assay results Choi et al. concluded that Cd(II)-(GSH)₂ is actively transported by ABCB6. Their main findings are that i) GSH is required for the transporter to interact with metal containing substrates and ii) ABCB6 can interact with various substrates due to the plasticity of its cavity and flexibility of NBDs.

Overall, the manuscript is well written and presented. The structural work is of solid quality, and so are most of the functional/biochemical assays. Further, the study appears well-embedded in the current "landscape" of other ABC transporters with similar function (metal transporters), and pertinent published work are cited extensively. We feel that the paper is a good match for Communication Biology in that it offers novel findings based on high quality experiments that are worth being reported. The comments below are meant to further improve the work.

Major concerns

1) The authors come to the conclusion that Cd(II)-(GSH)₂ is transported by ABCB6 without performing any transport assays at all. While stimulated ATPase activity in presence of GSH and Cd(II) are a strong indication that hABCB6 recognizes this substrates, it does not necessarily mean, that the complex is also transported. To support this statement, the authors should provide direct experimental evidence for transport, by performing transport assay as in their previous publication with coproporphyrin III. Alternatively, the authors should tone down their statements regarding GSH-Cd(II) transport-

2) The MST measurements to determine the binding affinity of the Cd(II)-(GSH)₂ complex to hABCB6 appear to be flawed. According to the materials and methods, the authors kept the GSH concentration and the hABCB6 concentration constant (1 μM GSH + 10 nM hABCB6) and added varying amounts of Cd(II) (0-62.5 mM Cd(II)). The range of Cd(II) concentrations were chosen ramping up to very high concentrations (in fact almost 7 orders of magnitude higher than the protein itself). In contrast, for getting the structure of the complex, the authors added 800 μM Cd(II) and 1 mM GSH. Can the authors explain why they performed the MST measurements this way? How can the authors exclude unspecific binding of Cd(II) to the transporter independently from GSH? How would the MST curves look like when taking a mutant that does no longer bind Cd(II)-GSH? Or a unrelated transporter not recognizing Cd(II)-GSH? Such controls would be important to further substantiate these experiments. Further, the reported data is also incomplete: 9/16 curves are missing from Fig. 1b and are also not in the supplementary material. Finally, the description of the data is contradictory at two levels. First, the MST measurement is presented as direct interaction between Cd(II) metal and protein in the text (line 111), but complex binding in the figure legend (line 420). Second, the direct interaction between Cd(II) and protein at the micromolar range contradicts with the overall conclusion of the paper, that GSH is required for interaction between ABCB6 and metal compounds.

Minor concern

1) Line 82: Chapter title: "Human ABCB6 actively exports Cd(II) in conjunction with GSH". No evidence was shown for active transport in the manuscript. Please rephrase.

2) Line 101: Cd(II):GSH complex stimulates, Cd(II) alone inhibits based on Fig. 1a.

3) Line 136-137: "... indicating a preferred state of the transporter under the given experimental conditions." Without ATP present, the transporter will not undergo major conformational changes.

4) Comparison of NBD tilts and separations, where the resolution is at best 5Å, inherently contains a large error. The low resolution in the NBD region suggests, that the protein is flexible. Such flexibility can be resolved with 3D variability analysis within CryoSPARC software.

5) The large number of particles in the final refinements (400'000) would allow for further classification of particles with 3D classification tools or 3D variability analysis in clustering mode in

CryoSPARC.

6) Typically membrane protein reconstructions are better with the non-uniform refinement algorithm. This would also help with the low resolution of the NBDs.

7) Line 185-187: "...alanine mutations of the GSH-interacting residues (R435A, R439A, R552A, N498A, Q501A, N545A, and T549A) led to a partial or complete loss of hABCB6core ATPase activity (Fig. 4e)."
Rephrase suggestion: mutations led to partial or complete loss of stimulated ATPase activity.

8) Comparison of ABCB6 capping residue (W546) with the plug in ABCB6 in G. Song et al, 2021, Cell Discovery, in both conformational states would be a valuable insight.

9) Lines 255-259: Unnecessary paraphrasing of last two sentences of the discussion. Did the authors use ChatGPT?

10) Line 305: GFP-resin washing step is missing from the protocol. Was it not performed?

11) Line 334: reference to ATPase assay protocol is wrong (#57). It points to a publication that did not have any ATPase assays.

Comments from Reviewers (Bold) and Author Responses

Reviewer #1 (Remarks to the Author):

The authors presented cryo-electron microscopy structures of core-human ABCB6 (ABCB6 Δ TMD0) bound to a cadmium Cd(II) ion in the presence of antioxidant thiol peptides glutathione (GSH) and phytochelatin 2 (PC2) at resolutions of 3.2 and 3.1 Å, respectively. The overall structure is inward-facing similar to the apo state. Two GSH molecules are symmetrically bound to the Cd(II) ion through their central cysteine, and biochemical assays delineate that presence of GSH associated metal ion effectively stimulates ATPase activity. Degrees of separation between the two halves of the transporter and tilt angles of the NBDs is substrate specific. The manuscript is overall well written and interesting.

Core hABCB6 is used in this paper as well as in previous papers, where TMD0 has been deleted i.e., the construct is Δ TMD0. This group has determined two structures of Cd(II)-bound hABCB6^{core} in complex with GSH or PC2. ABCB6 exhibits the differential requirement for GSH as a cofactor depending on the substrate type. GSH is metal lover. Wherever metal is there either in heavy metal state or porphyrin bound state, GSH facilitates/stimulates hydrolysis thereby resulting in transport.

Concerns:

(1) No transport assay is shown in this paper; however, transport assay was shown in the previous paper for Co-III only. This should be addressed.

>> To address this concern, we attempted to measure the Cd(II):GSH transport activity of hABCB6^{core} using liposome-entrapped fluorescent dyes (i.e., Leadmium green, Rhodamine B, and Fura-2) that are sensitive to changes in Cd(II) concentration. Unfortunately, despite our extensive efforts, meaningful data were not acquired. We hypothesized that Cd(II) ions might influence the stability and/or rigidity of liposomes, thereby impacting the transport activity of hABCB6^{core} (Kerek et al. *Biochimica et Biophysica Acta* 1858 (2016) 3169–3181).

>> Instead, we investigated whether overexpression of hABCB6^{core} could confer cellular resistance to cadmium ions. Cytotoxicity assays revealed that CdCl₂ in the culture medium kills Sf9 cells infected with recombinant baculoviruses carrying the catalytically inactive E752Q mutant at a concentration of 15 μ M. By comparison, cells infected with baculoviruses carrying the wild-type gene exhibited remarkable preservation of viability, even at a concentration as

high as 50 μM , reaching 40% relative to untreated controls. These results provide strong evidence that hABCB6^{core} protects cells from Cd(II)-induced cytotoxicity, utilizing energy derived from ATP hydrolysis. Detailed data are presented in Figure 1C and discussed in the revised manuscript (see text, lines 122–127).

(2) The degree of separation has a relation with ATPase activity, more the separation less the hydrolytic effectivity? is this correct?

>> In light of recent cryo-EM structures and biochemical analysis of ABCB6, it is clear that in the apo state, ABCB6 exhibits widely separated NBDs, leading to a low level of ATPase activity (*Protein Science* 2020; *Cell Discovery* 2021; *Mol Cells* 2022). Subsequent substrate binding induces a conformational change, bringing the NBDs into closer proximity, thereby enhancing ATPase activity. Hence, we propose that, similar to many other ABC transporters (though not applicable to all), ABCB6 follows the general rule that the degree of NBD separation correlates with ATPase activity.

(3) Previously Co-III-pPIX showed significant stimulation of hydrolysis in presence of GSH. Here Co(II) is used. Which is more biologically relevant? Free heavy metal ions or PPIX associated? This should be addressed.

>> We have addressed the suggested discussion on the potential oxidation state of cobalt ions and its biological relevance. Please see lines 274–279.

(4) Same is for cadmium in this study. Only free Cd(II) and Cd(II)-GSH were utilized. Is higher oxidation state possible here? Is there any possibility of complex formation with PPIX? Any biological relevance? These should be mentioned.

>> The proposed discussion has been addressed in the revised manuscript. Please see lines 281–284.

(5) Why not a structure in presence of non-hydrolysable ATP analog and metal ligands?

>> In a recent publication (*Communications Biology* 6:960, 2023), we reported that ABCB6 adopts either an occluded or outward-facing conformation in the presence of ADP·VO₄ and coproporphyrin III. We predict that this conformational preference will remain consistent even in the presence of ATP analog and metal ligands.

(6) How TM helix 7 is oriented here?

>> TM helix 7 is shown in Supplementary Figure 13.

(7) Fig 1(a)- For ATPase assay, hABCB6 protein concentration (800 μ M) and GSH concentration (1mM). Have you tried for different ratio like 1:2 or more?

>> To determine the optimal conditions for assessing the ATPase activity of ABCB6, various concentrations of Cd(II) (up to 900 μ M) and GSH (up to 5 mM) were tested in the presence of 1.4 μ M protein. The results showed that the experimental conditions yielding the highest ATPase activity contained 800 μ M Cd(II) and 1 mM GSH.

(8) Fig 1(c) - For normalized fluorescence change, what is the incubation time for GSH?

>> The incubation time (15 min) for GSH has already been mentioned in the Methods section. Please see lines 421–422.

(9) Fig (f) line 462, what is the significance of having conserved residues in 15 different species? Are these residues important for GSH:Cd(II) only? Not clear in the manuscript.

>> We have revised the text to clarify this point. Please see lines 209–220.

(10) All mutants gave partial or no binding except R552. The reasons are not explained properly. ATPase assay is not clear hence needs to be elaborated. ATP binding pocket has not been mentioned here.

>> It appears that the reviewer has misinterpreted the impact of the R552A mutation. We have revised the text to clearly state that all mutations, including R552A, led to partial or complete loss of Cd(II):GSH-stimulated ATPase activity of ABCB6 (lines 202–208).

>> We prefer not to discuss the ATP-binding pocket of ABCB6 here, as it has been discussed in detail in a previous study based on its structure in the ADP·VO₄-bound form (*Lee et al., Communications Biology 6:960, 2023*).

(11) line no.186: 'structural analysis, alanine mutations of the GSH-interacting residues (R435A, R439A, R552A, N498A, Q501A, N545A, and T549A) led to a partial or complete loss of hABCB6 core ATPase activity.' what about the residues of NBD? How does hydrolysis take place? What are the ATP interacting residues? Mutational studies in NBD will improve the manuscript.

>> As suggested, we generated the catalytically inactive E752Q mutant and conducted biochemical assays to evaluate its impact on ATP hydrolysis and cell protection from cytotoxicity, and to determine the binding affinity for Cd(II):GSH. Detailed data are presented in Figures 1c, 1d, and Supplementary Figure 1, and discussed in the revised manuscript (see lines 286–288).

(12) Time scan during measurement of ATPase activity would have been a better choice.

>> As suggested, the data has been reprocessed and is now presented in units of time. Please see Figures 1a, 1b, and 4e.

(13) Authors should discuss the about structural interface of NBD-TMD? Also, the paper <https://doi.org/10.1038/s41467-023-37851-9> should be referred.

>> We have cited the reference mentioned above and discussed the structural interface of NBD-TMD (lines 306–309). Detailed data are presented in Supplementary Figure 15.

(14) line no: 489- 'the different tilt angles and direction of the NBDs, which can influence the efficiency of ATP hydrolysis and subsequent transport processes, and are affected by the bound substrate species' ---- have they checked this with the other metal ions? Not clear.

>> In our experimental conditions, since ABCB6 only binds to Cd(II) in the presence of GSH, it is not possible to investigate how other metal ions affect the tilt angles of the NBDs.

(15) line no.76. 'Our structural findings shed light on the molecular mechanisms underlying ABCB6-mediated cadmium efflux'. The molecular mechanism of Cadmium efflux is not clearly understood.

>> In the revised manuscript, we have added a section discussing how ABCB6 facilitates cadmium efflux during the transport cycle. Please see lines 294–306 and Figure 7.

=====

Reviewer #2 (Remarks to the Author):

In the current study, Choi et al. show that cadmium binds to hABCB6 in a GSH-dependent manner and further determines the structure of cadmium-bound hABCB6 using cryo-EM and an hABCB6 mutant (ABCB6core) that they derived in a previous paper. ABCB6 is an important and current area of research. The protein may be critical for heme synthesis and is implicated in metal transport, transfusion compatibility, mitochondrial metabolism, xenobiotics, and cancer treatment. Despite the importance of ABCB6, its role in these processes is largely unstudied. Specific to this paper, the authors show that ABCB6 may play a role in cadmium toxicity and metabolism. This is an important advancement in the field of toxicology and continues their previous work with hABCB6 in porphyrin binding and transport. The authors do a wonderful job explaining the current research and made stunning figures. That said I would suggest the follow minor and possibly major revisions.

Possible major revision:

-The data set for the ATPase activity of hABCB6core + Cd(II) seems to be the same in figures 1a, 1d, and 4e; yet the negative control data sets (basal and GSH alone) in these figures all seem to be different. Was this a copying mistake? Were the same data sets used in different experiments? If this positive control data set was reused for different experiments, I would suggest that the authors repeat these experiments with the proper controls.

>> We appreciate the careful observation and concern raised by the reviewer. As suggested, the control experiments were independently repeated in each ATPase assay. While subtle differences were observed between datasets, we verified that the basal and Cd(II):GSH-stimulated activities were consistently maintained at similar levels. Based on these results, we have revised Figures 1a, 1d (now 1e), and 4e.

Minor revisions:

-For figures 1a, 1d, and 4e, I would suggest that the authors add statistical significance to their graphs, if any exists.

>> In response to the suggestion, we have indicated statistical significance (p -values) in

Figures 1a, 1d (now 1e), and 4e.

-It seems like from figure 1a, that GSH does increase the activity of hABCB6core. Was this significant or just a trend? This is different than what is stated in the text. I would suggest that the authors address this. Additionally, there seems to be a discrepancy between figures 1a and 1d. GSH in 1a seems to activate hABCB6core but has no effect in 1d. I would suggest that the authors address this.

>> It seems that the reviewer has misunderstood the effect of GSH on hABCB6^{core} activity. To avoid any potential confusion among readers, we have revised the figure legend to provide a clearer explanation of the observed effects.

Suggestions/Comments:

-Did cadmium in the absence of GSH inhibit hABCB6core activity? If so, I would suggest that the authors make a brief mention of this.

>> To evaluate the potential inhibitory effect of cadmium alone on hABCB6^{core} activity (Figure 1a), we conducted ATPase assays across various cadmium concentrations. As shown in Figure 1b, our results confirmed that there was no inhibitory effect of cadmium at any of the tested concentrations. Therefore, it is reasonable not to interpret the somewhat reduced ATPase activity of hABCB6^{core} in the presence of Cd(II) alone as significant.

-800uM metal concentration is quite high, and way beyond human physiological levels. Have the authors tried titrating the metals, and specifically the Cd(II) concentration?

>> We have tested various concentrations (up to 800 μ M) of metals to investigate their impact on stimulating the ATPase activity of hABCB6^{core}. Even at concentrations beyond physiological levels, only cadmium stimulated ABCB6 activity in conjunction with GSH. These data are presented in Figure 1b and discussed in the text (lines 115–119).

-Post-translational-modifications are seemingly very important for ABCB6. As the authors have noted, the precise location of ABCB6 is disputed and mature red blood cells seem to express glycosylated ABCB6 on the plasma membrane which may differ from mitochondrial ABCB6 if it exists in red cells. Given that the recombinant human ABCB6core used in this study is generated from Sf9 (armyworm cells), have the authors checked that the post-translational-modification profile of ABCB6core is similar to

ABCB6 found in human cells? The translational-value of these studies would be increased if this were known.

>> Previous studies demonstrated that post-translational modifications (e.g., N-glycosylation and disulfide bond formation) occur within the TMD0 region of ABCB6, NOT the core domain (*JBC*, 286, 8481-8492, 2011; *Biochem. J.* 467, 127–139, 2015). Therefore, cell type-specific expression is unlikely to affect the folding, activity, or stability of hABCB6^{core}. This point has been addressed in the revised manuscript (lines 45–48).

=====
Reviewer #3 (Remarks to the Author):

In this original research paper, Choi et al. report their newest findings on the human ABCB6 transporter. The authors determined the structure of nanodiscs-reconstituted, truncated hABCB6 in complex with Cd(II)-(GSH)₂ and Cd(II)-(phytochelatin)₂ in inward-facing conformations. Comparison of their new complex structures reported here to previously published substrate-bound structures of hABCB6 revealed large flexibility at the NBD region. To further support the physiological relevance of their structural findings, the authors performed in vitro ATPase assays in the absence and presence of substrates and determined the binding affinity of Cd(II)-(GSH)₂ to ABCB6 using microscale thermophoresis. Based on these assay results Choi et al. concluded that Cd(II)-(GSH)₂ is actively transported by ABCB6. Their main findings are that i) GSH is required for the transporter to interact with metal containing substrates and ii) ABCB6 can interact with various substrates due to the plasticity of its cavity and flexibility of NBDs.

Overall, the manuscript is well written and presented. The structural work is of solid quality, and so are most of the functional/biochemical assays. Further, the study appears well-embedded in the current “landscape” of other ABC transporters with similar function(metal transporters), and pertinent published work are cited extensively. We feel that the paper is a good match for *Communication Biology* in that it offers novel findings based on high quality experiments that are worth being reported. The comments below are meant to further improve the work.

Major concerns

1) The authors come to the conclusion that Cd(II)-(GSH)₂ is transported by ABCB6 without performing any transport assays at all. While stimulated ATPase activity in presence of GSH and Cd(II) are a strong indication that hABCB6 recognizes this

substrates, it does not necessarily mean, that the complex is also transported. To support this statement, the authors should provide direct experimental evidence for transport, by performing transport assay as in their previous publication with coproporphyrin III. Alternatively, the authors should tone down their statements regarding GSH-Cd(II) transport-

>> We have addressed this point above (Reviewer #1, comment #1).

2) The MST measurements to determine the binding affinity of the Cd(II)-(GSH)₂ complex to hABCB6 appear to be flawed. According to the materials and methods, the authors kept the GSH concentration and the hABCB6 concentration constant (1 μ M GSH + 10 nM hABCB6) and added varying amounts of Cd(II) (0-62.5 μ M Cd(II)). The range of Cd(II) concentrations were chosen ramping up to very high concentrations (in fact almost 7 orders of magnitude higher than the protein itself). In contrast, for getting the structure of the complex, the authors added 800 μ M Cd(II) and 1 mM GSH. Can the authors explain why they performed the MST measurements this way? How can the authors exclude unspecific binding of Cd(II) to the transporter independently from GSH? How would the MST curves look like when taking a mutant that does no longer bind Cd(II)-GSH? Or a unrelated transporter not recognizing Cd(II)-GSH? Such controls would be important to further substantiate these experiments.

>> To establish the optimal conditions for MST assays, we investigated various concentrations of protein, Cd(II), and GSH. However, concentrations >10 nM protein or >1 μ M GSH resulted in protein aggregation or its adsorption to the capillary, respectively, impeding accurate measurements.

>> Nevertheless, we understand the reviewer's concern about non-specific binding of Cd(II) to hABCB6^{core} under the current experimental conditions. To address this issue, we conducted additional MST assays using the hABCB6^{core}-Q501A mutant (incapable of Cd(II) binding) and hABCB10^{core} (unrelated to heavy metal transport) as negative controls. The results revealed little MST response from both controls when exposed to Cd(II):GSH, suggesting that the MST signal of hABCB6^{core} is positive. For further details, please refer to lines 135–137 and Figure 1d.

Further, the reported data is also incomplete: 9/16 curves are missing from Fig. 1b and are also not in the supplementary material.

>> We have corrected the figures related to the MST results. Please refer to Figure 1d and Supplementary Figure 2.

Finally, the description of the data is contradictory at two levels. First, the MST measurement is presented as direct interaction between Cd(II) metal and protein in the text (line 111), but complex binding in the figure legend (line 420). Second, the direct interaction between Cd(II) and protein at the micromolar range contradicts with the overall conclusion of the paper, that GSH is required for interaction between ABCB6 and metal compounds.

>> We have carefully revised the manuscript to remove any contradictions.

Minor concern

1) Line 82: Chapter title: “Human ABCB6 actively exports Cd(II) in conjunction with GSH”. No evidence was shown for active transport in the manuscript. Please rephrase.

>> As suggested, the subtitle has been changed to “Binding of cadmium to hABCB6^{core} significantly enhances ATPase activity in the presence of GSH”.

2) Line 101: Cd(II):GSH complex stimulates, Cd(II) alone inhibits based on Fig. 1a.

>> We have revised the text as suggested.

3) Line 136-137: “... indicating a preferred state of the transporter under the given experimental conditions.” Without ATP present, the transporter will not undergo major conformational changes.

>> The text has been revised. Please see lines 154–157.

4) Comparison of NBD tilts and separations, where the resolution is at best 5Å, inherently contains a large error. The low resolution in the NBD region suggests, that the protein is flexible. Such flexibility can be resolved with 3D variability analysis within CryoSPARC software.

>> As suggested, we conducted 3D variability analysis in cluster mode by employing a soft mask to exclude nanodisc density, along with three eigenvectors and a 5 Å low-pass filter. However, this approach did not yield significant improvements in the NBD map.

5) The large number of particles in the final refinements (400'000) would allow for further classification of particles with 3D classification tools or 3D variability analysis in clustering mode in CryoSPARC.

>> As suggested, we performed additional 3D classification using 400,000 particles to capture the most stable conformation of Cd(II):GSH-bound hABCB6^{core}. However, no further splits were observed.

6) Typically membrane protein reconstructions are better with the non-uniform refinement algorithm. This would also help with the low resolution of the NBDs.

>> We applied non-uniform refinement, but it was not significantly effective in improving the map quality for our data.

7) Line 185-187: "...alanine mutations of the GSH-interacting residues (R435A, R439A, R552A, N498A, Q501A, N545A, and T549A) led to a partial or complete loss of hABCB6core ATPase activity (Fig. 4e)." Rephrase suggestion: mutations led to partial or complete loss of stimulated ATPase activity.

>> We have revised the text as suggested.

8) Comparison of ABCB6 capping residue (W546) with the plug in ABCB6 in G. Song et al, 2021, Cell Discovery, in both conformational states would be a valuable insight.

>> In the revised version, we have addressed the suggested discussion on the structural changes of W546 and the TM7 loop (also known as plug) upon Cd(II):GSH binding. Please see lines 299–300 and Supplementary Figs. 13 and 14.

9) Lines 255-259: Unnecessary paraphrasing of last two sentences of the discussion. Did the authors use ChatGPT?

>> We have deleted the last sentence in the revised manuscript.

10) Line 305: GFP-resin washing step is missing from the protocol. Was it not performed?

>> We have revised the method description to include the omitted washing step for the GFP affinity resin. Please see lines 368–370.

11) Line 334: reference to ATPase assay protocol is wrong (#57). It points to a publication that did not have any ATPase assays.

>> This is our mistake. Thank you for pointing it out. We have corrected it.

REVIEWERS' COMMENTS:

Reviewer #1 (Remarks to the Author):

All previous concerns were adequately addressed and/or explained and the manuscript was significantly improved and strengthened. Now I recommend accepting it.

Reviewer #2 (Remarks to the Author):

The authors did well to address the reviewer comments and the publication is recommended for publication.

Reviewer #3 (Remarks to the Author):

The authors have addressed all my concerns. Of special note, the authors provide now additional MST control experiments which further substantiate their initially performed measurements.